# Generation and utilization of a HEK-293T murine GM-CSF expressing cell line

**Elektra Kantzari Robinson** [ID]◉, **Sergio Covarrubias** [ID]◉, **Simon Zhou** [ID]◉, **Susan Carpenter***

Department of Molecular, Cell and Developmental Biology, University of California Santa Cruz, Santa Cruz, CA, United States of America

◉ These authors contributed equally to this work.
* sucarpen@ucsc.edu

**Data Availability Statement:** Our vector maps are available in supplemental materials. The extended methods are also available on protocol.io (dx.doi.org/10.17504/protocols.io.bpkkmkuw). Our vectors and HEK-293T mGM-CSF cell line are

## Abstract

Macrophages and dendritic cells (DCs) are innate immune cells that play a key role in defense against pathogens. *In vitro* cultures of bone marrow-derived macrophages (BMDMs) and dendritic cells (BMDCs) are well-established and valuable methods for immunological studies. Typically, commercially available recombinant GM-CSF is utilized to generate BMDCs and is also used to culture alveolar macrophages. We have generated a new HEK-293T cell line expressing murine GM-CSF that secretes high levels of GM-CSF (~180 ng/ml) into complete media as an alternative to commercial GM-CSF. Differentiation of dendritic cells and expression of various markers were kinetically assessed using the GM-CSF HEK293T cell line, termed supGM-CSF and compared directly to purified commercial GMCSF. After 7–9 days of cell culture the supGM-CSF yielded twice as many viable cells compared to the commercial purified GM-CSF. In addition to differentiating BMDCs, the supGM-CSF can be utilized to culture functionally active alveolar macrophages. Collectively, our results show that supernatant from our GM-CSF HEK293T cell line supports the differentiation of mouse BMDCs or alveolar macrophage culturing, providing an economical alternative to purified GM-CSF.

## Introduction

Colony-stimulating factors (CSF) including macrophage colony-stimulating factor (M-CSF), granulocyte colony-stimulating factor (G-CSF), and granulocyte-macrophage colony stimulating factor (GM-CSF also known as colony stimulating factor 2, CSF2) are crucial for survival, proliferation, differentiation and functional activation of hematopoietic cells, including macrophages and dendritic cells (DCs) [1]. Macrophages and DCs are innate immune cells found in tissues and lymphoid organs that play a key role in defense against pathogens [2]. While there are a multitude of macrophage and dendritic cell subsets, GM-CSF is critical for the development of conventional dendritic cells (cDCs) and alveolar macrophages (AMs) [2]. Due to cell number limitations from harvesting cDCs and AMs directly from mice, well-established *in vitro* culturing of bone marrow and bronchoalveolar lavage fluid for dendritic cells and alveolar macrophages, respectively, using GM-CSF have become invaluable for immunological

available directly from the Carpenter lab email: sucarpen@ucsc.edu or can be requested from UCSC office of research via Jeffrey Jue, jljue@ucsc.edu, or innovation@ucsc.edu.

**Funding:** Work on this project was supported by R21AR070973 to Susan Carpenter, an award from the National Institute of Arthritis and Musculoskeletal and Skin Diseases (NIAMS). Additional support was obtained from startup funds from UCSC to the Carpenter lab. The funders had no role in study design, data collection and analysis, decision to publish, or preparation of the manuscript. Sergio Covarrubias received partial support from the R21AR070973 for his salary.

**Competing interests:** The authors have declared that no competing interests exist.

and molecular biology studies [2]. This has led to the use of CSF proteins in the purified form, as well as to the generation of recombinant cell lines that secrete the desired protein in the supernatant for cost efficiency [3–6]. One of the most widely adapted cell lines, utilized for differentiating and culturing murine bone-marrow derived macrophages (BMDMs) is the NCTC clone 929 strain L line, also known as L929 [7]. Supernatant from cultured L929 can be utilized, in lieu of purified M-CSF, to culture and differentiate BMDMs because it secretes murine M-CSF [7].

The Human embryonic kidney 293T (HEK293T) cell line is the ideal choice for expressing a CSF protein, for differentiating primary immune cells from mouse bone marrow, as it has been shown not to express innate immune pattern recognition receptors or naturally secrete immune-related cytokines [8, 9]. This ensures that only your protein of choice is predominantly expressed and that there is no inadvertent activation of the inflammatory cascade. Previously, murine GM-CSF has been cloned into J558L, a mouse B myeloma cell line [4], which is known to express cytokines, including IL10 [10]. Using J558L can therefore alter the results of *in vitro* experiments, by activating anti-inflammatory pathways when cultured with BMDCs or alveolar macrophages.

In this study we constructed HEK293T cell lines stably expressing and secreting murine GM-CSF. We utilized GM-CSF to generate and culture BMDCs and alveolar macrophages (AMs). HEK293T cells have no expression of human GM-CSF and the constructed cell lines, thus express only the stably transfected murine GM-CSF. We have found that our line is very stable, producing GM-CSF at a concentration of ~200 ng/ml even following freeze thaw cycles. We differentiated BMDCs and cultured AMs using our supGM-CSF and compared them to commercially available purified GM-CSF (pGM-CSF) and found that our supGM-CSF yields a higher number of cells, purity of DCs is not altered, and the cells show intact immune signaling cascades.

## Materials and methods

### Cloning strategy of mGM-CSF

The mGM-CSF gene, including PspXI and NotI restriction sites, was amplified from the pCR3.1-mGM-CSF vector (Addgene, 74465). The PCR was set up with: 25 µl 2X Phusion High-Fidelity PCR Master Mix (Thermo Scientific), 1 µl PspXI_hMCSF_fwd (20mM): TCCGCTCGAGCCACCATGTGGCTGCAGAATTTACTTTTCC, 1 µl NotI_hMCSF_rev (20mM): GACGCGGCCGCTCATTTTTGGCCTGGTTTTTTGC, 1 µl pCR3.1-mGM-CSF (20ng) and 22 µl DEP-C nuclease-free water. PCR program: 95°C 3 minutes, 35 cycles of 95°C 30 sec, 60°C 30 sec, and 72°C 1 min, and end the PCR with 72°C and 12°C hold. The PCR product was purified using the PCR Clean-up Kit (Macherey-Nagel) and was subsequently digested with PspXI (New England BioLabs) and NotI-HF (New England BioLabs) using recommended digestion conditions (https://nebcloner.neb.com/#!/redigest) and was cloned into our custom pSico bidirectional lentiviral vector (sequence in supplemental). Sequence was confirmed by Sequetech (Sanger) sequencing.

### Lentivirus generation

HEK-293T (ATCC CRL-3216) cells (4e5 cells/well) were plated onto a 6-well plate (353046, Corning) with complete DMEM [10% heat-inactivated FCS (Gibco, 26140–079), 100 µg/ml penicillin (Thermo, 15140122), and 100 µg/ml streptomycin (Thermo, 15140122)]. 24 hr later, 500 ng of pSico-mGM-CSF or empty vector control, 250 ng psPAX2 (Addgene, 12260), and 250ng pMD2.G (Addgene, 12259) were mixed in 200µl of serum-free Opti-MEM (Gibco) and 5µl Lipofectamine 2000 (Thermo Fisher) was added to mix and incubated for 20 minutes at

room temperature. Transfection reaction was added to HEK-293T cells and allowed to transfect for 72 hr, and supernatant was harvested, passed through 0.45μm filters (Millipore, Stericup), and aliquots were stored at -80˚C.

## Construction of GM-CSF-producing HEK-293T Cells

HEK-293T cells were transduced 2 days with 200 μl of lentivirus per 1e5 cells. 48 hr after infection, the HEK-293T cells were selected with puromycin (2 μg/ml) for >3 days, monitoring viability and increase of mCherry expression using FACS (Attune NxT Flow Cytometer). Once mCherry expression exceeded 85%, puromycin was removed, and cell-line was expanded and used to produce GM-CSF. A detailed method for GM-CSF supernatant production is described in S2 File.

## Enrichment of DCs and macrophages from BM and in-vitro stimulation

Bone marrow (BM) cells were harvested from the femurs and tibia of wild-type C57BL/6 between 6- and 18-weeks old mice and depleted of erythrocytes using red blood cell (RBC) lysis (Biosciences 786–649). 1e6-3e6 BM cells were plated per well of a 6-well tissue culture plate (353046, Corning) in complete DMEM supplemented with either 10% M-CSF from L929 cells, 5–10% supernatant from the mGM-CSF 293T cell line, or 10–25 ng/ml of recombinant mGM-CSF [PMC2015, Thermo Fisher]. Media was replaced on day 3 and every 2 days henceforward. Cells were scraped and moved onto a larger plate as they proliferated. After 7 to 14 days of enrichment, cells were stimulated by adding 200 μg/ml LPS (Sigma, L2630-10MG) to the media and harvested for analysis after 0, 6, or 24 hr of stimulation. Total cell count was determined by staining cells with trypan blue and using a Light Microscope and hemocytometer.

## Western blot

Undiluted supernatant was loaded from .9–3.6 ng, which was quantified by ELISA. 1μg of recombinant GM-CSF (Cat#576302) was used as a positive control. Each sample was resolved by SDS-PAGE and transferred to a polyvinylidene difluoride (PVDF) membrane and Western blotted with either mGM-CSF antibody (1:500, R&D DY415-05) and horseradish peroxidase-conjugated b-actin monoclonal antibody (1:5,000, Santa Cruz Biotechnology) were used as a loading control. HRP-conjugated streptavidin (1:60, R&D DY415-05) secondary antibodies were used.

## Surface staining for flow cytometry

Stimulated BMDCs and BMDMs were harvested and centrifuged at 300 g for 5 minutes. The pellets were resuspended in 100 μl sorting media (2% FCS, 5 mM EDTA, 1X PBS). Each sample was blocked with 100 μl of Fc block CD16/CD32 (BD Biosciences) diluted 1:250 in sorting media and incubated at room temperature for 15 minutes. Antibodies and viability dye were then used to stain the samples for 30 minutes on ice in the dark. Antibodies and dye: APC-eFluor 780 anti-CD45 (Invitrogen, 47-0451-82), Alexa-Fluor 700 anti-CD11c (BD Pharmingen, 560583), FITC anti-CD11b (Thermo Fisher, MA1-10081), PE-eFluor 610 anti-F4/80 (eBioScience, 61-4801-80), PE anti-MHC Class II (BD Pharmingen, 562010), and Fixable Aqua Dead Cell Stain (Thermo Fisher, L34957). Cells were washed twice in 1 ml sorting media, spinning at 300 g for 5 minutes. The resulting pellet was resuspended in 200 μl of sorting media and analyzed by FACS (Attune NxT Flow Cytometer).

## Inflammasome activation assay and ELISA

supGM-CSF (25 ng/ml) or recGM-CSF (25 ng/ml) differentiated BMDCs were plated at 2e5cells/well in a 96 well plate with complete DMEM media. Cells were first primed with 200 ng/ml LPS for 3 hr prior to treatment with different agents. Poly(dA:dT) DNA (dsDNA mimic) was transfected using Lipofectamine 2000 at a concentration of 1.5 μg/ml, 6 hr prior to harvesting supernatant. The supernatant was harvested using a p300 multichannel pipette and stored at -80˚C. Commercial ELISAs were used to measure the following proteins in mice in triplicate following the manufacturer's instructions: IL6 (mouse IL6 DuoSet, R&D Systems DY406), IL1β (Mouse IL-1β/IL-1F2 R&D Systems DY401-05) and mGM-CSF (Mouse GM-CSF DuoSet Systems DY415-05).

## BALF harvest and alveolar macrophage culturing

Bronchoalveolar Lavage Fluid (BALF) was harvested as previously stated by Cloonan *et al.* (PMID:26752519). 40 mice were euthanized by $CO_2$ narcosis, the tracheas cannulated, and the lungs lavaged with 0.5-ml increments of ice-cold PBS eight times (4 ml total), samples were combined in 50 ml conical tubes. BALF was centrifuged at 500 *g* for 5 min. 1 ml red blood cell lysis buffer (Sigma-Aldrich) was added to the cell pellet and left on ice for 5 min followed by centrifugation at 500 *g* for 5 min. The cell pellet was resuspended in 500 μl PBS, and leukocytes were counted using a hemocytometer. Specifically, 10 μl was removed for cell counting (performed in triplicate) using a hemocytometer. Cells were plated in sterile 12 well plates at 5e5/well (total of 8 wells) and use complete DMEM with 25 ng/ml supGM-CSF.

## Culturing of alveolar macrophages

24 hr post-BALF isolation, media was removed and fresh complete DMEM with 25 ng/ml supGM-CSF is added. All cells that adhere to the surface of the plate are considered alveolar macrophages (AM) as previously determined by Chen *et al.* [11]. After new media is added, AMs are stimulated with 200 ng/ml LPS (Sigma, L2630-10MG). Harvest supernatant 6 hr post-stimulation. Harvested supernatant was sent to Eve technologies for cytokine analysis. Statistics were performed using GraphPad prism.

## Cytotoxicity assay

Cytotoxicity was assessed by measuring the release of LDH into the media (LDH-Cytotoxicity Colorimetric Assay Kit II; BioVision) according to the manufacturer's protocol.

## Ethics statement

All animal work was carried out in strict accordance with the recommendations in the Guide for the Care and Use of Laboratory Animals of the National Institutes of Health. The protocol was approved by the Institutional Animal Care and Use Committee at the University of California Santa Cruz (Protocol number CARPS1810).

# Results

## Production of a murine GM-CSF-secreting HEK-293T cell line

A murine GM-CSF (mGM-CSF) containing plasmid (Addgene, 74465) was used as a template to amplify mGM-CSF, which we inserted into a plasmid with a pSico backbone (S1 Fig, S1 File). The vector contains a bidirectional EF1a promoter driving puromycin resistance gene and mCherry on one side and mGM-CSF on the other (S1B Fig, Fig 1A). Lentivirus containing

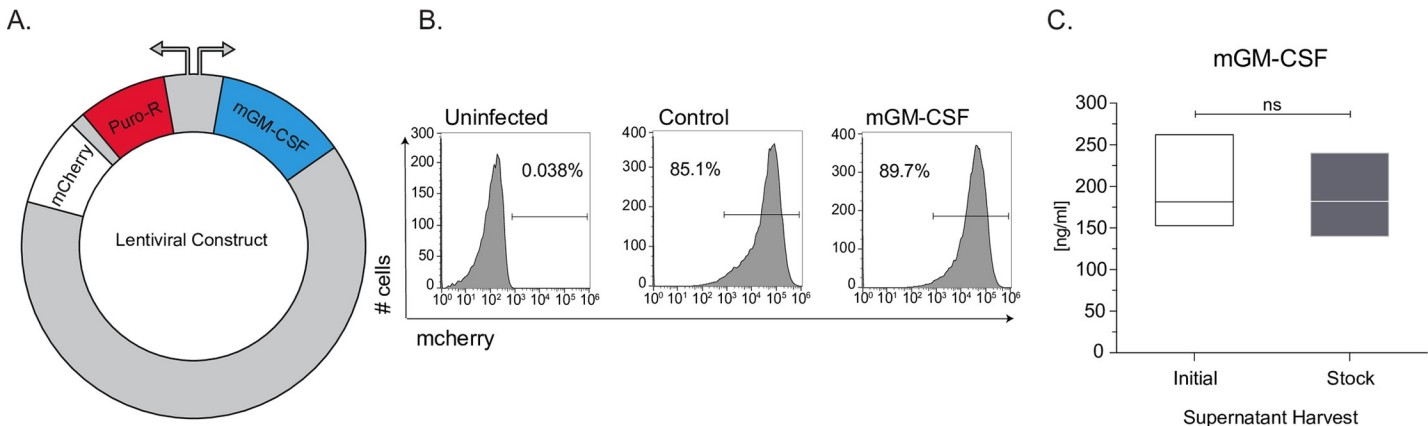

**Fig 1. Generation of mouse GM-CSF secreting HEK293T cell line.** (A) Schematic of lentiviral plasmid construct used to generate the stable HEK293T mGM-CSF expressing cell line. (B) Uninfected, Control plasmid and mGM-CSF plasmid lentivirally infected HEK239T cells were puromycin selected for mCherry expression and assessed using flow cytometry. (C) Protein secretion and cell-line stability of mGM-CSF HEK293T cells was confirmed ELISA, using initial and stock cell-line. Error bars represent the standard deviation of biological triplicates. Student's t-tests were performed using GraphPad Prism. (SN) indicates not significantly different.

mGM-CSF, or a control construct was generated and transduced into HEK293Ts and monitored using flow cytometry (S2 Fig). Post puromycin selection for 7 days, our control and mGM-CSF constructs were incorporated into the genomes of HEK293T cells (at a rate of >85%) (Fig 1B). In order to determine how much GM-CSF was being produced by the HEK293T line we plated and grew $2 \times 10^6$ cells for 3 days, isolated the supernatant and performed an ELISA to measure the concentration which was found to be 181.7 ng/ml (S2 File). Additionally, GM-CSF protein size was assessed by western blot showing it is running as a single band at ~17kDa (S3 Fig). The cell line was subsequently frozen down into aliquots and thawed to assess stability. Protein secretion was assessed again, by ELISA, and the stock secreted mGM-CSF at 181.9±ng/ml (Fig 1C). There was no significant difference between the concentration of mGM-CSF in the supernatant of the initial or the freeze thawed HEK293T cell line indicating that the line is stable.

## Bone-marrow cells differentiated with supGM-CSF yield a higher number of dendritic cells compared to pGM-CSF

To assess the efficacy of the mGM-CSF-rich supernatant (supGM-CSF), bone marrow (BM) cells were treated with 10 ng/ml or 25 ng/ml of supGM-CSF and compared to the same concentrations of a commercially obtained purified GM-CSF (pGM-CSF) [12–14]. As a control for differentiation, we also generated bone-marrow derived macrophages (BMDMs) using supM-CSF obtained from L929 cells [15] (Fig 2A).

Cells incubated with M-CSF, pGM-CSF and supGM-CSF were fully differentiated by day 7 and could be cultured until day 14 (Fig 2A). Morphologically, BM differentiated with supGM-CSF and pGM-CSF, at either 10 ng/ml or 25 ng/ml, have a more stellate morphology as is expected in dendritic cells compared to M-CSF differentiated macrophage cells. There is no morphological difference between pGM-CSF or supGM-CSF differentiated cells, at either 10 ng/ml or 25 ng/ml (Fig 2B). At day 9 of differentiation, cells were harvested and counted. When comparing the pGM-CSF to the supGM-CSF, supGM-CSF yields significantly more viable cells in comparison to pGM-CSF (Fig 2C). After culture with M-CSF, pGM-CSF or supGM-CSF, cells were assessed for purity by flow cytometry based on previously published panels [16, 17]. After gating on Live+/CD45+/CD11b+ cells purity was determined based on the proportion of the population expressing CD11c and F4/80, shown as a quadrant

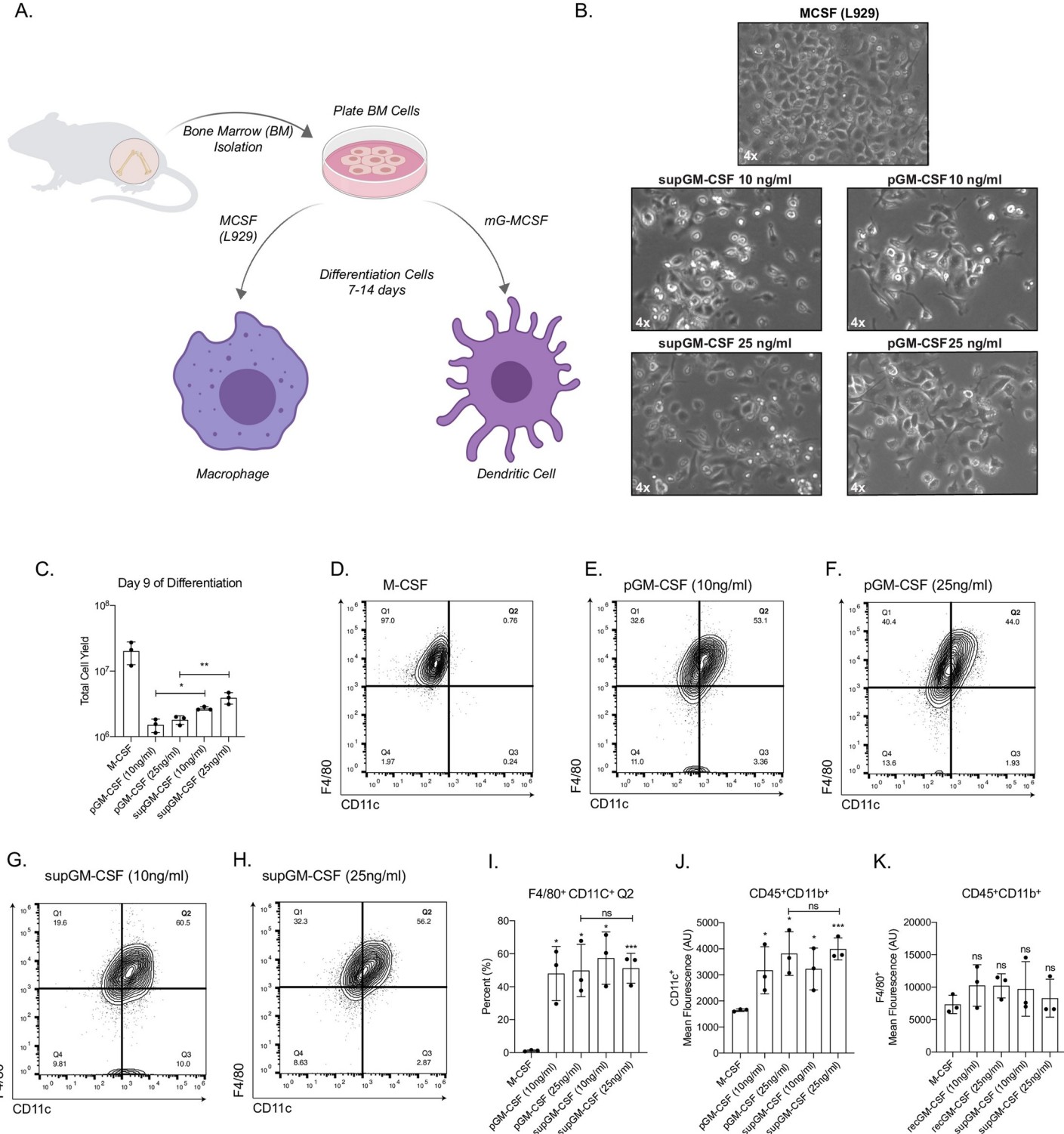

**Fig 2. Generation of primary dendritic cells using purified GM-CSF or supernatant GM-CSF. (A)** Schematic of differentiation of primary bone marrow-derived macrophages (BMMs) or primary bone marrow-derived dendritic cells (BMDCs). **(B)** Images of primary BMMs and BMDCs under a light microscope after 7 days of differentiation. **(C)** Total number of live cells generated from using one of 5 conditions: M-CSF, purified GM-CSF (pGM-CSF) at 10 ng/ml, pGM-CSF at 25 ng/ml, supernatant GM-CSF (supGM-CSF) at 10 ng/ml or supGM-CSF at 25 ng/ml. **(D)** Gating of CD45+ and CD11b+ M-CSF differentiated cells are represented in a quadrant contour plot to determine the expression of F4/80 and CD11c. Same gating strategy is used for **(E)** pGM-CSF at 10 ng/ml, **(F)** pGM-CSF at 25 ng/ml, **(G)** supGM-CSF at 10 ng/ml and **(H)** supGM-CSF at 25 ng/ml. **(I)** Graphical representation for the percentage of M-CSF or GM-CSF differentiated cells that are F4/80+ and CD11c+ (Q2). **(J)** Graphical representation for the mean fluorescence of CD11c+. **(K)** Graphical representation for the mean fluorescence of F4/80+. Error bars

represent the standard deviation of n = 5. Student's t-tests were performed using GraphPad Prism. Asterisks indicate statistically significant differences between mouse lines (*p $\geq$ 0.05, **p $\geq$ 0.01, ***p $\geq$ 0.005).

(S4 Fig, Fig 2D–2H). Percent of cells in quadrant 2 (F4/80+ and CD11c+) was significantly higher for all GM-CSF treated cells, in comparison to M-CSF treated cells, as is expected since BMDMs express very little CD11c while BMDCs highly express CD11c. However, pGM-CSF and supGM-CSF were not significantly different (Fig 2I). As expected, the mean fluorescence intensity for dendritic cell marker CD11c+ cells were significantly higher for all GM-CSF treated cells in comparison to M-CSF (Fig 2J), while F4/80+ expression was equal for every treatment (Fig 2K). While supGM-CSF is able to generate more viable cells than pGM-CSF, there is no significant difference in cellular purity between supGM-CSF and pGM-CSF (both at ~85% purity).

## BMDCs generated using supGM-CSF function as efficiently as BMDCs generated using pGM-CSF

Dendritic cells are the professional antigen presenting cells of the immune system. In order to test the ability of our supGM-CSF to generate functional DCs we measured their levels of MHC-Class II (MHCII) expression using flow cytometry [18]. M-CSF differentiated cells express little MHCII, while pGM-CSF and supGM-CSF express robust levels of MHCII, when differentiated with 10 ng/ml or 25ng/ml concentration of GM-CSF (Fig 3A and 3B, S4A and S4B Fig). Dendritic cells differentiated with 10 ng/ml GM-CSF show statistically higher mean fluorescence intensity for MHCII for supGM-CSF in comparison to pGM-CSF (S5C Fig). Interestingly, dendritic cells differentiated with 25 ng/ml of either pGM-CSF or supGM-CSF exhibit no significant difference in overall MHCII protein expression when evaluated by mean fluorescence intensity (Fig 3C). On a more granular level, supGM-CSF appears to have a single population expressing high levels of MHCII in comparison to the two populations of cells generated by the pGM-CSF expressing low and high MHCII (Fig 3A and 3B). In addition to assessing MHCII levels, we also assessed the ability of the pGM-CSF and supGM-CSF BMDCs to respond to inflammatory stimulation. Both pGM-CSF and supGM-CSF BMDCs were stimulated with LPS (200 ng/ml) for the indicated time course and IL6 mRNA expression was measured by quantitative PCR (qPCR) (Fig 3C). Thus, determining that the supGM-CSF derived cells maintain their inflammatory activity. Finally, AIM2 inflammasome activity was measured through IL1b protein secretion of both purified and supernatant GM-CSF, while priming of the system with LPS alone was assessed by IL6 secretion. Here we observe that supGM-CSF produces significantly higher expression of IL6 at a protein level, but not transcriptionally (Fig 3D and 3E) suggesting that these dendritic cells are more sensitive to inflammatory activation compared to DCs generated using pGM-CSF.

## supGM-CSF can maintain viable and inflammatory inducible alveolar macrophages

mGM-CSF is a critical protein factor that is not only necessary for driving primary dendritic cell differentiation, but also for maintaining primary alveolar macrophages (AM) in culture [19, 20]. To test the ability of supGM-CSF to maintain AM in culture, we harvested bronchiolar lavage fluid (BALF) from 40 WT wild-type mice. We pooled these cells, counted and then plated them in 25 ng/ml supGM-CSF supplemented media. 24 hr post-harvest, using a light microscope, we tested that AMs attached (Fig 4). Viability of AMs were assessed by measuring the amount of lactate dehydrogenase (LDH) in the media. The amount of measured LDH correlates directly with the cell number lysed [21]. Our data indicate that supGM-CSF does not negatively affect the cell culture, while the controls indicate there are healthy cultured cells not

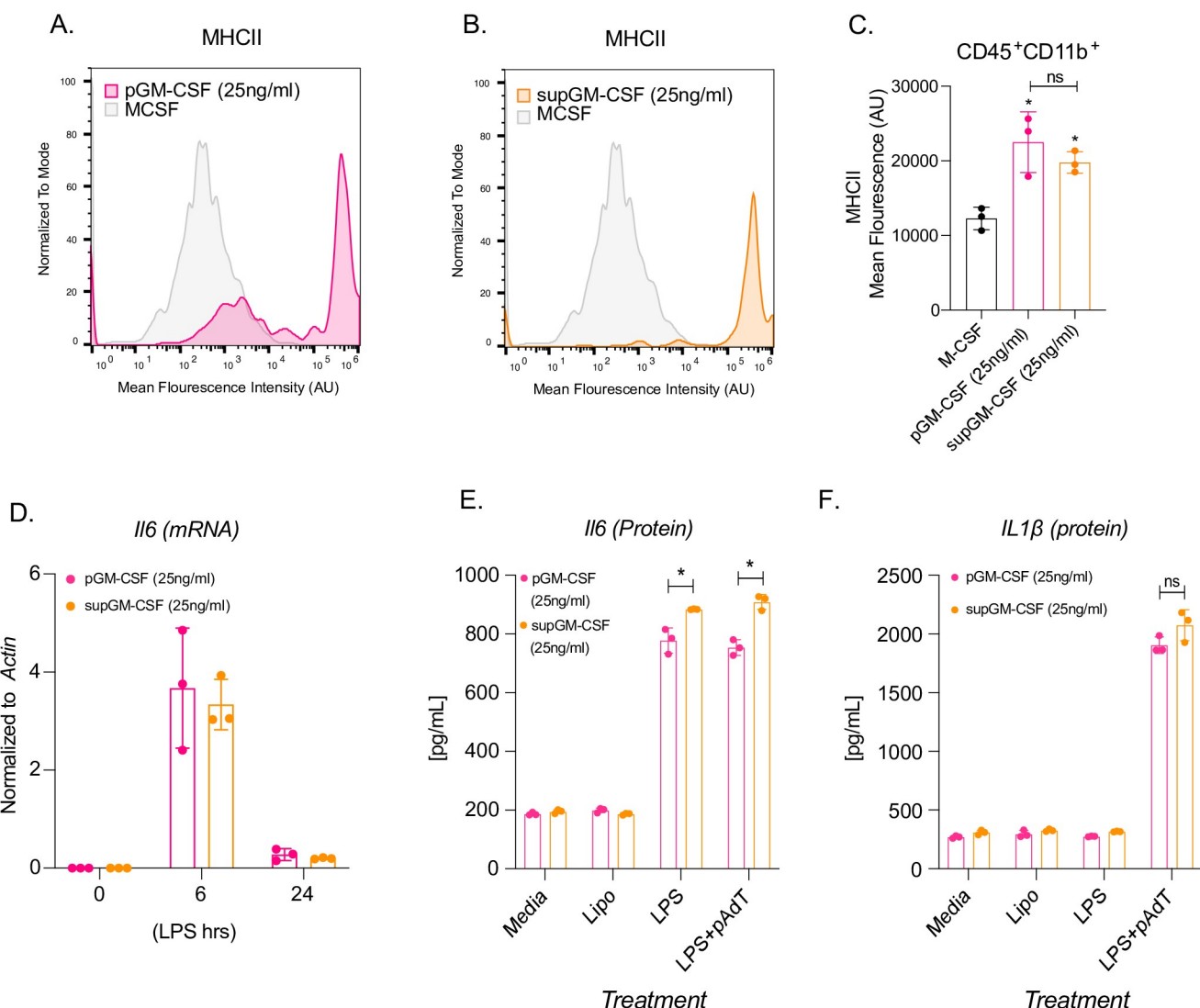

**Fig 3. Dendritic cells differentiated with supGM-CSF retain dendritic cell activity and are more inflammatory in comparison to pGM-CSF. (A)** Histogram overlay of CD45+ and CD11b+ M-CSF (grey) and pGM-CSF (pink) or **(B)** supGM-CSF (orange) differentiated cells expressing MHCII. **(C)** Graphical representation of MHCII mean fluorescence intensity (MFI) of M-CSF, pGM-CSF, or supGM-CSF differentiated cells. **(D)** Il6 transcript measure by RTq-PCR from pGM-CSF (pink) or supGM-CSF (orange) DCs stimulated with LPS for 0, 6, and 24 hr. (E) Dendritic cells differentiated with pGM-CSF (pink) or sup-GM-CSF (orange) were treated with lipofectamine, LPS (24 hr) or LPS (2 hr) and polydA:dT (6 hr) secreted IL-6 or IL-1β **(E)** was measured by ELISA. All experiments were performed in biological triplicate. Student's t-tests were performed using GraphPad Prism. Asterisks indicate statistically significant differences between mouse lines ($^*$p $\geq$ 0.05, $^{**}$p $\geq$ 0.01, $^{***}$p $\geq$ 0.005).

resistant to apoptosis (Fig 4B). AMs were stimulated with LPS for 24 hr and cytokines were measured by ELISA. Inflammatory inducible proteins are significantly upregulated in AMs cultured in supGM-CSF media, including proteins IL6, TNFa, MDC, MCP-1, IP10, KC and Rantes (Fig 4C–4I). supGM-CSF cultured primary alveolar macrophages maintain their pro-inflammatory activation programming.

## Discussion

Dendritic cells and macrophages have been cultured and studied for the last 40 years, leading to many advances in culturing protocols. Flt3L is often utilized as a factor for BMDC

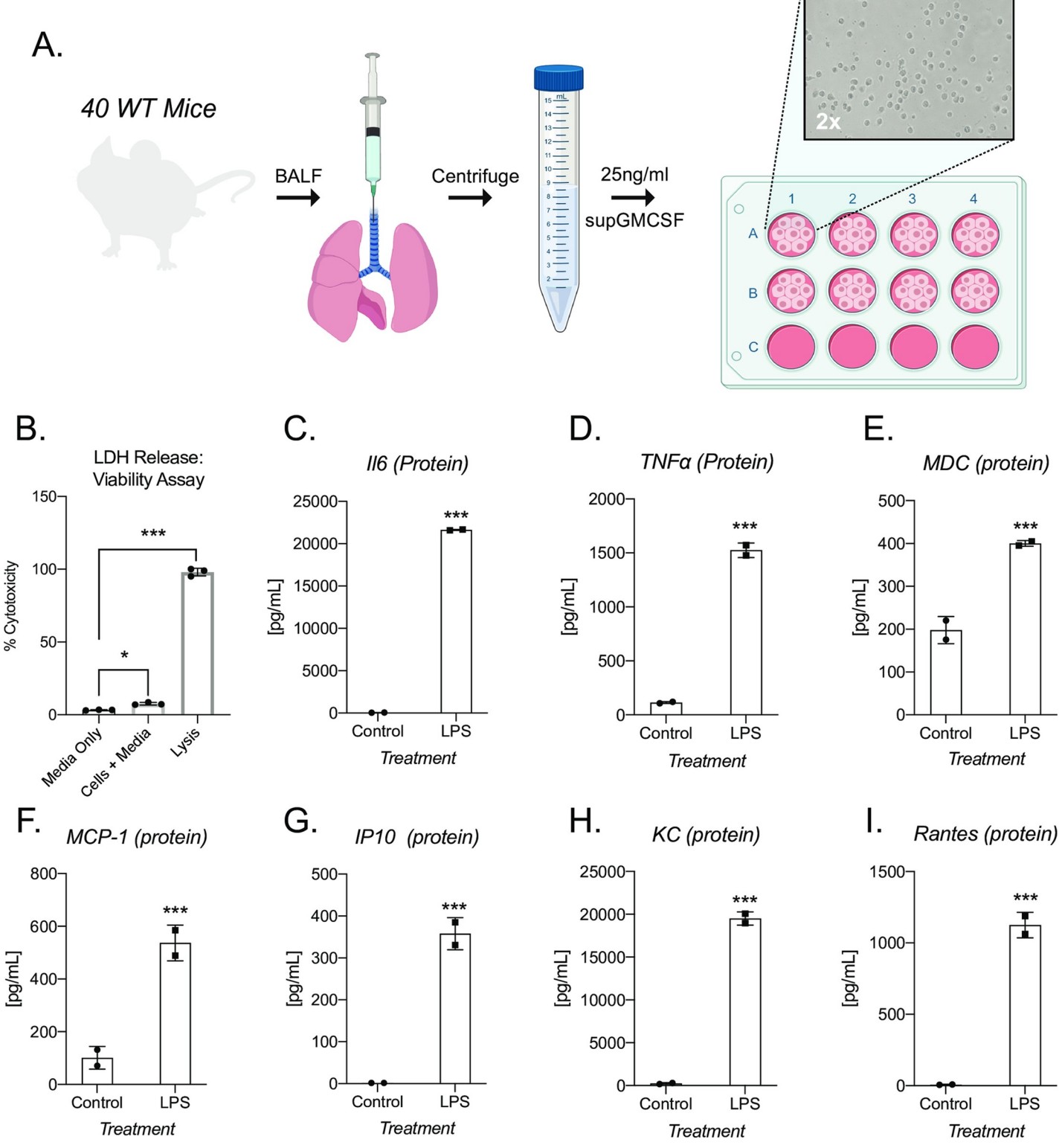

**Fig 4. supGM-CSF can differentiate functional alveolar macrophages.** (**A**) Schematic of alveolar macrophage supGM-CSF harvesting and differentiating experiment. (**B**) The cellular viability of differentiated alveolar macrophages was measured by LDH, where media, cells, and lysis of cells were measured. (**C**) Secreted IL6 protein was measured from supGM-CSF differentiated alveolar macrophages treated with and without LPS for 6hr, as well as (**D**) TNFa, (**E**) MDC, (**F**) MCP-1, (**G**) IP10, (**H**) KC and (**I**) Rantes. All experiments were performed in biological triplicate. Student's t-tests were performed using GraphPad Prism. Asterisks indicate statistically significant differences between mouse lines (*p $\geq$ 0.05, **p $\geq$ 0.01, ***p $\geq$ 0.005).

differentiation, it is now appreciated that Flt3L DCs are representative of steady-state resident DCs, while GM-CSF BMDCs mirror the transcriptional programing of pro-inflammatory recruited cells [22, 23]. Granulocyte-macrophage colony-stimulating factor (GM-CSF), differentiated bone marrow cells are widely used as a model system for conventional DC development [24, 25], as well as sustaining primary alveolar macrophages in culture [26]. Current strategies for the generation of murine BMDCs or culturing of primary alveolar macrophages utilize store bought purified recombinant GM-CSF or a stable cell-line expressing recGM-CSF called J558L [10]. J558L, a murine GM-CSF, secreting cell-line is utilized for both BMDC and alveolar macrophage culturing, however it is an immune cell that also secretes IL-10 which can alter the transcriptional programing of the cells when culturing and only secretes 80 ng/ml [10, 19, 20].

Using supernatant from GM-CSF secreting HEK293T cells, can serve as an alternative to purified GM-CSF for murine BMDCs or maintenance of primary alveolar macrophages. We have successfully cloned and stably integrated murine GM-CSF into HEK293T cells, which consistently secrete 180 ng/ml of GM-CSF (Fig 1). GM-CSF protein size is predicted to be ~16 kDa. Our sup-GM-CSF runs as a single band in the supernatant at approximately 17 kDa. Purified GM-CSF obtained from biolegend runs at a smaller size of ~14.5 kDa. The size differential could be due to the way in which the recombinant protein is produced in *E.Coli* or that the protein produced from our cell lines contains modifications that result in it running at a higher molecular weight. Bone marrow differentiated with our supGM-CSF produces more cells by day 9 compared to pGM-CSF, but both GM-CSF sources generate an equal percentage of pure dendritic cells based on previously published gating strategies [16, 17] (Fig 1C and 1I). Additionally, using more GM-CSF also will produce more viable and proliferating cells, which can be further enhanced with higher concentrations of supGM-CSF (Fig 2C). Commercial GM-CSF can be used instead of our HEK293T supGM-CSF, but it is expensive, and GM-CSF has to be added every 2 days to cells during the differentiation process. Our supGM-CSF is much more cost effective compared to purified GM-CSF from a number of companies (S1 Table). From one harvest of supGM-CSF you can generate ~10 μg, which will be the cost of 50 ml of complete media (S2 File), while 10 μg of purified GM-CSF from Thermo Fisher will cost $276. The purity of BMDCs generated from pGM-CSF and supGM-CSF does not differ when assessed by flow cytometry based on previously published panels [16, 17] (S3 Fig, Fig 2D–2H).

DCs are professional antigen presenting cells and therefore express high levels of MHC Class II [22, 27]. BMDCs differentiated with either Flt3L or GM-CSF have comparable T cell activation, and therefore MHC II expression [23]. BMDCs differentiated using 25 ng/ml with either pGM-CSF or supGM-CSF express MHC-II at a comparable level (Fig 3C), when measured by mean fluorescence intensity (MFI) through flow cytometry, while on a more granule level it appears that the supGM-CSF has a purer population of cell expressing high MHCII compared to the pGM-CSF which has two distinct populations (Fig 3A and 3B). Interestingly, BMDCs differentiated with 10 ng/ml GM-CSF show statistically higher MFI for MHCII for supGM-CSF in comparison to pGM-CSF (S4A–S4C Fig). Based on previously reported data from Helft *et al.* we hypothesize that our supGM-CSF differentiates pure BMDCs because of the MHCII high expression, while the pGM-CSF generates a mixed population of BMDCs and BMDMs represented by the MHCII intermediate and high populations [28].

As previously stated, picking a cell line to express a recombinant protein is very important. If an immune cell is chosen, one risks the chance of having immune factors being secreted in the supernatant, leading to priming and activation of either pro- or anti- inflammatory transcriptional programs [10, 19, 20]. By performing a time-course stimulation or overnight LPS stimulation, our study indicates supGM-CSF does not inhibit the inflammatory program transcriptionally when measuring the transcript or protein level of IL6 (Fig 3D and 3E).

supGM-CSF and pGM-CSF differentiated BMDCs both retain the ability to activate the inflammasome pathway, leading to the secretion of IL1b (Fig 3D–3F). While there has been published reports that it is macrophages within the population of cells that arise from the GM-CSF model of differentiation and not the DCs that are responsible for the inflammasome activation we can show here that our CD11c[+]/MHC[hi] DC cells possess a functional AIM2 inflammasome [29].

Not only does our study provide data to support the use of supGM-CSF compared to pGM-CSF for BMDCs, but we also show that supGM-CSF can be utilized to sustain the culturing of primary alveolar macrophages from murine bronchoalveolar lavage fluid (BALF) [10, 19]. After 48 hr of supGM-CSF cultured BALF, the adhered alveolar macrophages (AMs) were >95% viable (Fig 4B). More importantly, when stimulating the cells with LPS overnight cytokines such as IL6, TNFa, KC and Rantes were all inducible (Fig 4C–4I). Thus, supGM-CSF maintains AMs in culture and does not inhibit the pro-inflammatory programming of the immune cells.

Taken together our results show that a pure BMDC population and inflammatory inducible DCs or AMs can be established by culturing BM cells or BALF with a crude supernatant from our GM-CSF HEK293T cell line. This line will provide the research community with a more cost-effective alternative to commercially available GM-CSF.

## Supporting information

**S1 Fig. Validation of cloned murine GM-CSF. (A)** Image of PspXI and NotI digested PCR product of murine GM-CSF. **(B)** Detailed 266 vector map of mGM-CSF, restriction sites, promoter, antibiotic selection and mCherry sequence. **(C)** Alignment of Sanger sequencing results of mGM-CSF and 266 Vector map strategy. **(D)** Quality of sanger sequencing results of murine GM-CSF.
(TIF)

**S2 Fig. mCherry expression in HEK293T cell line.** HEK-293T cell line lentiviral integration of GM-CSF was selected using puromycin (2 μg/ml) over a week. Flow cytometry was used to assess fluorescence intensity.
(TIF)

**S3 Fig. Size quantification of pGM-CSF and supGM-CSF.** Western blot from left to right: 1 μg of pGM-CSF, Ladder, 20ul of 293T supernatant, 5 μl of supGM-CSF (0.9 ng), 10 μl (1.8 ng) of supGM-CSF, 15 μl (2.7 ng) of supGM-CSF and 20 μl (3.6 ng) of supGM-CSF.
(TIF)

**S4 Fig. Gating strategy for identifying dendritic cell and macrophage populations.** M-CSF and GM-CSF differentiated cells were both put through the same gating strategy. The 5 gating plots are for the **(A)** M-CSF, **(B)** pGM-CSF (10 ng/ml), **(C)** pGM-CSF (25 ng/ml), **(D)** supGM-CSF (10 ng/ml), **(E)** supGM-CSF (25 ng/ml) differentiated cells. Non-debris cells were gated for in gate 1, then singlets in gate 2, followed by CD45+ and live cells were gated for in gate 3, then CD11b+ cells were gated in gate 4 finally this population of cells were visualized using F4/80+ and CD11c+ markers and gate were put into quadrants.
(TIF)

**S5 Fig. Expression of MHC II on dendritic cell populations differentiated with 10ng/ml pGM-CSF or supGM-CSF. (A)** Histogram overlay of CD45+ and CD11b+ M-CSF (grey) and pGM-CSF (pink) or **(B)** supGM-CSF (orange) differentiated cells expressing MHC II. **(C)** Graphical representation of MHC II mean fluorescence of M-CSF, pGM-CSF, or supGM-CSF

differentiated cells. Student's t-tests were performed using GraphPad Prism. Asterisks indicate statistically significant differences between mouse lines ($^{*}$p $\geq$ 0.05, $^{**}$p $\geq$ 0.01, $^{***}$p $\geq$ 0.005). (TIF)

**S1 Table. Cost of purified murine GM-CSF.**
(TIF)

**S1 File.**
(APE)

**S2 File. Extended methods.**
(DOCX)

**S1 Raw Images.**
(PDF)

## Author Contributions

**Conceptualization:** Sergio Covarrubias, Susan Carpenter.

**Data curation:** Elektra Kantzari Robinson, Simon Zhou.

**Formal analysis:** Elektra Kantzari Robinson.

**Funding acquisition:** Susan Carpenter.

**Methodology:** Elektra Kantzari Robinson, Sergio Covarrubias, Simon Zhou.

**Project administration:** Sergio Covarrubias.

**Supervision:** Sergio Covarrubias, Susan Carpenter.

**Visualization:** Elektra Kantzari Robinson.

**Writing – original draft:** Elektra Kantzari Robinson.

**Writing – review & editing:** Elektra Kantzari Robinson, Sergio Covarrubias, Simon Zhou, Susan Carpenter.

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
