## [Decision Letter · Decision Letter 0]

6 Oct 2020

PONE-D-20-18789

Generation and utilization of a HEK-293T murine GM-CSF expressing cell line

PLOS ONE

Dear Dr. Carpenter,

Thank you for submitting your manuscript to PLOS ONE. After careful consideration, we feel that it has merit but does not fully meet PLOS ONE’s publication criteria as it currently stands. Therefore, we invite you to submit a revised version of the manuscript that addresses the points raised during the review process.

The reviewers request a numbers of clarifications in the manuscript text, and verification that the GM-CSF protein expressed from the transgene is of the expected size and was not fused to a cellular gene during the integration process. To ensure rigor and reproducibility, please clarify whether the data presented represent multiple independent experiments or replicates from a single experiments. Please double-check the statistical analyses, for example the data presented in Fig. 3F do not appear to be significantly different but are marked as so. Please add statistical analyses to Fig. 4B-I.

We look forward to receiving your revised manuscript.

Kind regards,

Donna A. MacDuff, Ph.D

Academic Editor

PLOS ONE

Journal Requirements:

2. Thank you for including your ethics statement: "In our work, we used mice for all experiments. These procedures were approved by IACUC, our number is CARPS1810.".

i) Please amend your current ethics statement to include the full name of the ethics committee that approved your specific study.

ii) Once you have amended this/these statement(s) in the Methods section of the manuscript, please add the same text to the “Ethics Statement” field of the submission form (via “Edit Submission”).

For additional information about PLOS ONE submissions requirements for ethics oversight of animal work, please refer to http://journals.plos.org/plosone/s/submission-guidelines#loc-animal-research

3. Thank you for stating the following financial disclosure: 'SCa received R21AR070973 funding for this work.'

**Please include your amended statements within your cover letter; we will change the online submission form on your behalf.**

Reviewers' comments:

Reviewer's Responses to Questions

**Comments to the Author**

1. Is the manuscript technically sound, and do the data support the conclusions?

Reviewer #1: Yes

Reviewer #2: Yes

2. Has the statistical analysis been performed appropriately and rigorously? 

Reviewer #1: Yes

Reviewer #2: Yes

3. Have the authors made all data underlying the findings in their manuscript fully available?

Reviewer #1: Yes

Reviewer #2: Yes

4. Is the manuscript presented in an intelligible fashion and written in standard English?

Reviewer #1: Yes

Reviewer #2: Yes

5. Review Comments to the Author

Reviewer #1: In this manuscript, Robinson et al describe a new cell line that they have developed that expresses constitutively GM-CSF. This 293T derived cell line that expresses high levels of GM-CSF in the media, is a cheap and efficient alternate source of GM-CSF that is usually bought commercially at high cost. They provide an array of experiments that show their supGM-CSF is as potent in the differentiation of BMDCs and AMs as the commercially acquired GM-CSF. Furthermore, they show that the BMDCs and AMs developed show no changes in their immune signaling cascades. Overall, this is a very interesting study and this cell line could make the study of BMDCs and AMs more affordable and easier to do.

Points

1. As the authors are using a lentivirus to introduce the GM-CSF into the 293T cells. Do they know where the lentivirus integrated? Did they aim for a single integration or multiple integrations? If the lentivirus integrated in multiple sites, this may affect the stability of the line.

2. The authors should run a western blot showing that GM-CSF is at the proper molecular weight. This is particularly important as they transduced the 293T cells with a lentivirus. ELISAs do not tell you if the secreted GM-CSF is intact or fused with a protein at the site of the lentiviral intergration. Even if it is functionally active, showing that the GM-CSF secreted is at the right molecular weight and not fused with some cellular protein is of importance.

3. Line 102 “the following …mice” makes no sense. Please fix this part of the sentence

4. It is not clear in the figure legends or the text if the repeats of the BMDCs/BMDMs experiments (staining, response to LPS etc) were done using cells differentiated from one mouse or each repeat is BMDCs/DMs derived from a different mouse every time (i.e. cells from different mice are used for different repeats of the experiments). Please clarify.

5. In line 178 you say you use 10WT mice, but Figure 4A says 40. Which one is right?

Reviewer #2: The authors have generated and characterized a cell line that produces and secretes GM-CSF. This reagent may be useful to the field for use in lieu of purified recombinant GM-CSF to generate bone-marrow derived dendritic cells. This manuscript is suitable for publication in Plos ONE, and I have only a few comments/suggestions.

Line 80: “…depleted of erythrocytes.” How was this done? Lysis buffer or other method?

Line 81-82: “…supplemented with either 10% M-CSF from L929 cells, 5-10% supernatant, or 10-25ng/ml of recombinant mGM-CSF” To what cells does “5-10% supernatant” refer? I presume it is the 293T cells, but this should be clarified here.

Line 97: I suggest removing or replacing “inflammasome” in this method title because the activation of these cells includes LPS stimulation and does not appear to be specific to the inflammasome.

Line 117: add citation for “Chen et al. (PMID:3288696)” to references

Line 153: “…CD11c is a dendritic cell marker” CD11c is also expressed by some macrophages.

Line 164-165: “Dendritic cells differentiated with 10ng/ml GM-CSF show statistically higher mean fluorescence intensity for MHC class II for supGM-CSF in comparison to pGM-CSF”. The authors should cite Helft et al. (PMID: 26084029), which show that MHCII high population are more DC-like and the MHC-intermediate population is more macrophage-like. The differences noted by the authors in their comparisons may relate to these two distinct populations reported by Helft et al. to be present within the GM-CSF cultures.

Line 165-167: Although there is no difference in MFI overall, there appears to be a difference in proportion of MHCII-low/int cells between Fig. 3A and Fig. 3B.

Line 221: “…while this pathway is now contested in DCs our results for GM-CSF is true” The meaning of this statement is unclear.

Line 372: This extended method does not appear to be finalized. For example, there are ?’s remaining in the method text (line 381, 426, 436)

6. PLOS authors have the option to publish the peer review history of their article (what does this mean?). If published, this will include your full peer review and any attached files.

Reviewer #1: No

Reviewer #2: No

---

## [Author Response · Author response to Decision Letter 0]

17 Dec 2020

See response to reviewers letter attached.

---

## [Editor Report · Decision Letter 1]

9 Feb 2021

PONE-D-20-18789R1

Generation and utilization of a HEK-293T murine GM-CSF expressing cell line

PLOS ONE

Dear Dr. Carpenter,

Thank you for submitting your manuscript to PLOS ONE. After careful consideration, we feel that it has merit but does not fully meet PLOS ONE’s publication criteria as it currently stands. Therefore, we invite you to submit a revised version of the manuscript that addresses the points raised during the review process.

Although you have addressed the reviewers' concerns, a question has arisen regarding adherence to PLoS ONE publication criteria for studies that describe new methods and tools as indicated here:

https://journals.plos.org/plosone/s/submission-guidelines

Specifically:

"Utility: The tool must be of use to the community and must present a proven advantage over existing alternatives, where applicable."

"Validation: If similar options already exist, the submitted manuscript must demonstrate that the new tool is an improvement over existing options in some way. This requirement may be met by including a proof-of-principle experiment or analysis; if this is not possible, a discussion of the possible application and some preliminary analysis may be sufficient".

Conditioned media from your new cell line is more cost-effective than commercially available rGM-CSF. However, an alternative cell line, J558L-GM-CSF, is available and still in use today for the purposes described in your study, despite the 1997 study that indicated possible detrimental secretion of IL-10. Such studies include:

https://www.sciencedirect.com/science/article/pii/S0092867417313880

https://bio-protocol.org/e3302 (reference 19)

https://www.sciencedirect.com/science/article/pii/S1931312818305419?via%3Dihub

Do you have experimental evidence that indicates that your cell lines produces conditioned media that is superior to that produced by the J558L cells line for the purpose of culturing BMDCs or alveolar macrophages? Is there evidence in the literature that J558L cell supernatant is problematic for the in vitro differentiation and culturing of the cell types used in your study?

"Availability: If the primary focus of a manuscript is the presentation of a new tool... it should be openly available under a license no more restrictive than CC BY." 

Will your 293T-GM-CSF cell line be made available to other investigators? If so, please include this information in the Data and Materials Availability section (lines 303-306).

"Cell lines: For established cell lines, the Methods section should include: ... The cell line repository or company the cell line was obtained from, the catalogue number, and whether the cell line was obtained directly from the repository/company or from another laboratory."

Please include this information in the Methods section. It will be particularly important for a reagent that may be used by other laboratories in the future. 

I apologize for the delay in returning this decision to you.

We look forward to receiving your revised manuscript.

Kind regards,

Donna A. MacDuff, Ph.D

Academic Editor

PLOS ONE

---

## [Author Response · Author response to Decision Letter 1]

9 Mar 2021

Dear Editor,

We appreciate your time and continued interest in our manuscript. We believe we have addressed all the concerns to the best of our abilities. Below we outline our specific responses to the four main comments/concerns and kept the quoted submission guidelines in red. Overall we continue to believe that this work has the scientific merit to be published in PLOS One.

Editor Comments 

Submission guidelines: "Utility: The tool must be of use to the community and must present a proven advantage over existing alternatives, where applicable."

Submission guidelines: "Validation: If similar options already exist, the submitted manuscript must demonstrate that the new tool is an improvement over existing options in some way. This requirement may be met by including a proof-of-principle experiment or analysis; if this is not possible, a discussion of the possible application and some preliminary analysis may be sufficient".

1. Conditioned media from your new cell line is more cost-effective than commercially available rGM-CSF. However, an alternative cell line, J558L-GM-CSF, is available and still in use today for the purposes described in your study, despite the 1997 study that indicated possible detrimental secretion of IL-10. Such studies include:

Our tool, HEK-293T mGM-CSF cells is much more advantageous than the existing alternatives. When assessing the main alternative, recombinant GM-CSF, our cell line is much more economical as well as more efficient (Fig. 2C). The recombinant GM-CSF is the gold standard when it comes to both culturing bone marrow-derived dendritic cells (BMDCs) (https://doi.org/10.1002/cpim.115) and alveolar macrophages (10.1084/jem.20131199, 10.1189/jlb.1107781, 10.1007/978-1-4939-6625-7_23). While J558L, a myeloma, mGM-CSF cell line exists and was created in 1988 (https://doi.org/10.1002/eji.1830180115), it has not replaced recombinant mGM-CSF when it comes to general in vitro studies. This is primarily due to the fact that the cell line is derived from a lymphoid (immune cell) and therefore can secrete more than just mGM-CSF into the supernatant from this cell line, including the incredibly important anti-inflammatory cytokine IL10 found in the 1997 study (10.1002/eji.1830270326). When this supernatant is then cultured with bone marrow to differentiate into BMDCs or fetal livers to differentiate alveolar macrophages, for instance, it will affect the transcriptional programming (immune signaling) leading to a switch of a pro-inflammatory to an anti-inflammatory state. Our cell line, HEK-293T mGM-CSF is much more advantageous because these cells are considered “empty” in that they do not express these immune chemokines and will not alter the state of immune cells.

From publicly available RNAseq datasets (https://www.proteinatlas.org/) [image below], we know that IL10 is not expressed in HEK-293T cell lines. This, in addition to how the cell line was generated and the cell type, makes our HEK-293T mGM-CSF cell line superior to the J558L mGM-CSF lymphoid expressing cell line.

At this moment, the only additional cytokine that is secreted in J558L and not HEK-293Ts is IL10. IL10 is a heavily studied cytokine, defined as a potent anti-inflammatory that plays a central role in limiting host immune response to pathogens, thereby preventing damage to the host and maintaining normal tissue homeostasis (PMID: 22428854). In DCs a study from the Gregori group showed that IL10 can lead to a very specific subset of DCs, coined DC-10 in humans (10.3389/fimmu.2018.00682). Another study, by the Staege group shows that IL10 changes both the phenotype and gene expression of BMDCs (PMID: 24222115). 

Additionally, papers that were indicated as sources that use J558L can be misleading. This is because J558, originally generated in 1972 (10.1073/pnas.80.3.825), was further subcultured and a J558L cloned was identified and defined as a spontaneous heavy chain-loss-variant myeloma cell line and does not express GM-CSF (10.1073/pnas.80.3.825). 

Now, we would now like to discuss the papers you cited. First, the Cell study by the Colonna group shows the use of J558L specifically as a cancer cell line (https://doi.org/10.1016/j.cell.2017.11.037). In this study, J558L was utilized as a cancer cell line to show the anti-cancer properties of NK cells and PDGF-DD with the presence of pro-inflammatory cytokines (TNFa or IFNg). This study is utilizing J558L which does not express GM-CSF, which again is being primarily used as a myeloma lymphoid cancer cell line to determine novel anticancer mechanisms. The cell line was not utilized to generate Alveolar Macrophages or Dendritic Cells. Therefore this use does not discount the value of our HEK-293T mGM-CSF cell line. Second, the Cell Host & Microbe study by the Nice group does utilize J588L mGM-CSF to generate DCs (https://doi.org/10.1016/j.chom.2018.10.003). In this study the entirety of the study was not in vitro, however, I would be very concerned about how a potent anti-inflammatory protein would affect the cytokine profile of a primary BMDC. This would be a huge caveat to the experiment. We understand the costliness of rGM-CSF and this may outweigh the experiment caveats if the proper controls are used. However, we believe that our tool would be greatly advantageous to all researchers performing in vitro studies on BMDCs and Alveolar Macrophages with an interest in pro-inflammatory signaling. The final paper that was cited by you the editor was a 2019 study, our reference 19, showing the use of J558L mGM-CSF to culture alveolar macrophages (10.21769/BioProtoc.3302) from bronchiolar lavage (BAL) fluid. This protocol’s main aim is to determine the best way to harvest the highest number of alveolar macrophages by assessing temperature and FCS or EDTA concentration in the media used to perform BALs. The cultured primary alveolar macrophages were then counted, the activity of these cells was never measured. Thus the caveat of the IL-10 expression was not considered as critical and J558L was used instead of the recombinant GM-CSF standard. Therefore in conclusion we believe our tool is worth publishing in PLOS One and available as the new gold standard that will be a cheaper and overall more efficient alternative to rGM-CSF and does not come with the caveats of inflammatory cytokines as are associated with the J558L.

2. Do you have experimental evidence that indicates that your cell lines produce conditioned media that is superior to that produced by the J558L cell line for the purpose of culturing BMDCs or alveolar macrophages? Is there evidence in the literature that J558L cell supernatant is problematic for the in vitro differentiation and culturing of the cell types used in your study?

We were primarily interested in making sure that our cell line was more advantageous when compared to the gold-standard of BMDC or Alveolar Macrophage in vitro work. We chose to focus on making the comparison of our line to the most utilized alternative in the field with his recombinant GM-CSF. Given the caveats outlined above with the J558 line producing cytokines like IL10, we did not consider it to be the best comparison to make as it is will be clear to immunologists interested in studying signaling that J558L is just not an option for use in such studies. We also already know that the J588L cell line expresses immune proteins, including the anti-inflammatory protein IL10 (10.1002/eji.1830270326). 

Submission guidelines: "Availability: If the primary focus of a manuscript is the presentation of a new tool... it should be openly available under a license no more restrictive than CC BY." 

3. Will your 293T-GM-CSF cell line be made available to other investigators? If so, please include this information in the Data and Materials Availability section (lines 303-306).

This is a very important point. We would like to share that this cell line is available upon request. Since being published in bioRxiv, July 2020, our cell line has been sent to 4 separate labs for both culturing BMDCs and primary Alveolar Macrophages. The ‘Data and Materials Availability’ section has been amended (lines 305-306). 

Submission guidelines: "Cell lines: For established cell lines, the Methods section should include: ... The cell line repository or company the cell line was obtained from, the catalogue number, and whether the cell line was obtained directly from the repository/company or from another laboratory."

4. Please include this information in the Methods section. It will be particularly important for a reagent that may be used by other laboratories in the future. 

Material and Methods amended for HEK-293T cell line, line 70.

---

## [Editor Report · Decision Letter 2]

12 Mar 2021

Generation and utilization of a HEK-293T murine GM-CSF expressing cell line

PONE-D-20-18789R2

Dear Dr. Carpenter,

We’re pleased to inform you that your manuscript has been judged scientifically suitable for publication and will be formally accepted for publication once it meets all outstanding technical requirements.

Kind regards,

Donna A. MacDuff, Ph.D

Academic Editor

PLOS ONE
---

## [Editor Report · Acceptance letter]

31 Mar 2021

PONE-D-20-18789R2 

Generation and utilization of a HEK-293T murine GM-CSF expressing cell line 

Dear Dr. Carpenter:

I'm pleased to inform you that your manuscript has been deemed suitable for publication in PLOS ONE. Congratulations! Your manuscript is now with our production department. 

Kind regards, 

on behalf of

Dr. Donna A. MacDuff 

Academic Editor

PLOS ONE